# DESIGN PRINCIPLES FOR TD-BASED MULTI-POLICY MORL IN INFINITE HORIZONS

## ABSTRACT

Multi-objective reinforcement learning (MORL) addresses problems with multiple, often conflicting goals by seeking a set of trade-off policies rather than a single solution. Existing approaches that learn many policies at once have shown promise in deep settings, but they depend on supervised retraining and carefully curated data, making them ill-suited for online and infinite-horizon tasks. Temporal-difference (TD) methods offer a natural alternative, as they update policies incrementally during interaction, but current TD-based approaches are limited to small, episodic problems. In this work, we present design principles for extending TD-based multi-policy MORL to both predictable (stationary) and flexible (non-stationary) policies, to avoid spurious dominance relations, and to detect cycles. Through ablation studies, we show how each principle contributes to recovering diverse and reliable policies, providing a principled path toward scalable TD-based multi-policy methods in deep MORL.

## 1 INTRODUCTION

Multi-objective reinforcement learning (MORL) is an RL setting with multiple, often conflicting objectives. MORL algorithms aim to approximate the Pareto front (PF) — the set of value vectors (and corresponding policies) for which no objective can be improved without degrading another — capturing trade-offs to support preference-based decision-making (Hayes et al., 2021). In standard RL, no single policy can accommodate all preference trade-offs; objectives are typically aggregated into a single scalar reward function via scalarization (combining multiple rewards into one) (Roijers et al., 2013). When preferences are unknown beforehand, scalarization is insufficient, as it requires retraining for each preference and often provides limited PF coverage (Van Moffaert & Nowé, 2014). Such challenges motivate methods that learn multiple policies to represent the full preference space.

Approaches for learning multiple policies fall into two categories. Outer-loop methods iteratively derive new policies from previously learned ones (Roijers et al., 2015; Parisi et al., 2017; Röpke et al., 2024), while inner-loop methods aim to learn the entire policy set in a single run (Van Moffaert & Nowé, 2014; Ruiz-Montiel et al., 2017; Reymond & Nowé, 2019; Reymond et al., 2022). Pareto Conditioned Networks (PCNs) (Reymond et al., 2022) extend inner-loop methods to deep RL by conditioning policies on desired return vectors, enabling scalability to high-dimensional settings. However, PCNs are not based on temporal-difference (TD) updates and rely on costly supervised-style retraining and careful trajectory curation, which limits scalability and online adaptation for infinite-horizon tasks. TD methods, in contrast, learn incrementally at each interaction and naturally support infinite-horizon settings. Yet, existing TD-based multi-policy approaches remain largely restricted to tabular, episodic settings (Van Moffaert & Nowé, 2014; Ruiz-Montiel et al., 2017).

To apply tabular inner-loop methods to the infinite-horizon setting via TD learning, we introduce a series of design choices to tackle the challenges of this setting: allow tracking and following policies when multiple options exist; learning solutions for tasks requiring predictable behavior vs. discovering a broader range of solutions; ensuring policies are not prematurely discarded; computing undiscounted returns to give the user an accurate sense of performance when selecting the policy; and detecting cycles, which offer well-defined returns for infinite horizons. Together, these elements provide the basis for a systematic analysis of tabular multi-policy RL in infinite-horizon settings, providing a principled foundation for future deep-RL extensions.

**Problem Setting:** MORL is essential for tasks with conflicting objectives. Multi-policy methods like PCNs can produce solution sets in deep RL but rely on costly supervised retraining and trajectory curation, limiting scalability and online adaptation in infinite-horizon settings. TD methods support incremental online learning but remain mostly tabular and episodic. As a result, no TD-based multi-policy approach effectively handles infinite-horizon MORL, forcing a trade-off between supervised and simplistic TD methods. We address this gap by adapting tabular TD inner-loop methods to infinite horizon problems, laying a principled foundation for future deep-RL extensions.

**Our contributions include:** **(i)** A systematic analysis of tabular multi-policy MORL in infinite-horizon settings, outlining design principles guiding TD-based deep-RL methods. **(ii)** A novel trajectory-based framework to track and execute both stationary and non-stationary policies, handling cycles and supporting robust policy selection and trade-off analysis by the user. **(iii)** Identification and resolution of spurious domination from reward horizon mismatches, reward estimate frequency, and incomplete structural information, with mechanisms for managing trajectories and cycles efficiently, supporting reliable learning and retrieval of policies.

## 2 RELATED WORK ON MORL

MORL seeks to optimize several, typically conflicting, objectives, with preferences often unknown beforehand. The goal is to find a set of non-dominated policies, where no objective can be improved without sacrificing another. Such policies represent trade-offs subject to preference rather than an objectively best solution, forming what is known as the Pareto front (PF) (Hayes et al., 2021).

Multi-objective problems can be converted into single-objective ones through *scalarization*; however, choosing weights is difficult, and even exhaustive searches often miss parts of the PF where non-isomorphic weight–objective mappings cause similar weights to yield very different trade-offs (Das & Dennis, 1997; Van Moffaert & Nowé, 2014). Inspired by Reward-Free Exploration (RFE) (Jin et al., 2020), where agents first explore (learning) and later optimize for any reward formulation (planning), Preference-Free Exploration (PFE) (Wu et al., 2020) allows agents to learn with rewards and then search based on specific preferences. However, PFE also relies on weights to define the preference and requires a new search for every new preference set. Multi-policy methods approximate the PF directly, allowing users to analyze trade-offs without predefined preference weights. Outer-loop approaches (Roijers et al., 2015; Parisi et al., 2017; Röpke et al., 2024) iteratively derive new policies from those learned in previous runs. In contrast, inner-loop methods (Van Moffaert & Nowé, 2014; Ruiz-Montiel et al., 2017; Mandow & de-la Cruz, 2018; Reymond & Nowé, 2019; Reymond et al., 2022; 2024), learn the entire policy set simultaneously within a single run.

In tabular approaches, Van Moffaert & Nowé (2014) keeps track of policies by separating returns into average immediate and future discounted rewards, while Ruiz-Montiel et al. (2017); Mandow & de-la Cruz (2018) links each policy to the next action in a state. These methods are limited to episodic, acyclic settings, restricting use in infinite-horizon problems. In deep RL, early single-run multi-policy work (Reymond & Nowé, 2019) struggled to scale even in simple domains. More recently, Reymond et al. (2022) introduced Pareto Conditioned Networks (PCNs), training a supervised model on an iteratively updated trajectory dataset to produce actions conditioned on a target return and horizon. While PCNs yield flexible policies, they inherit key limitations of supervised/imitation-style training: costly periodic retraining, adaptation constrained by dataset refresh rate, and reliance on careful trajectory curation and exploration. In contrast, temporal-difference (TD) methods update policies online at each interaction, using reward feedback for immediate improvement and more reliable credit assignment — crucial in infinite-horizon settings. In this paper, we extend inner loop tabular methods to infinite-horizon settings, laying a foundation for TD deep RL extensions. For a broader MORL survey, see Hayes et al. (2021).

## 3 METHODOLOGY

We introduce a TD-based multi-policy MORL framework (see Figure 1 for a summary of components) that adds *colors/labels* to transitions (Section 3.1) and structures them into stationary or non-stationary trajectories (Section 3.2). Our method detects what we call *spurious dominations* (Section 3.3) and enables policies with well-defined undiscounted returns in extended infinite-horizon

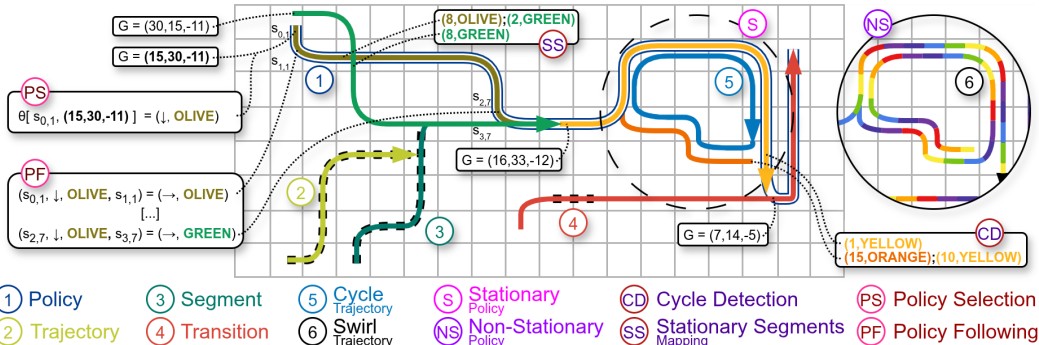

Figure 1: **Overview of the components underpinning our analysis.** A uniquely-colored segment forest: a policy (1) unfolds as a sequence of trajectories (2), each decomposed into segments (3) made of transitions (4). Trajectories may form cycles (5) or swirls (6), and policies can be stationary (S) or non-stationary (NS). We highlight cycle detection (CD), the Stationary Segments Mapping (SS), and the policy selection (PS) and policy following (PF).

settings (Section 3.4). It further provides explicit tracking of stationary segments (Section 3.5) and cycle detection (Section 3.6), ensuring stationarity.

## 3.1 FOLLOWING A POLICY THROUGH COLORED TRAJECTORIES

In RL, a policy $\pi$ guides the agent along trajectories $\tau$ (each trajectory is a sequence of transitions $\delta$), mapping states to action distributions that maximize cumulative reward. Since the agent can start from any state, a policy can also be seen as a forest of independent or converging trajectories. In multi-policy settings, multiple non-dominated solutions may exist per state-action (e.g., in Figure 2, $(s_{00}, \rightarrow)$ leads to two different solutions), producing a set of non-dominated trajectories and making it non-trivial to determine which action continues a policy. Without careful tracking, the agent may deviate from its intended trajectory and follow a dominated policy, even if each action it takes remains non-dominated (Van Moffaert & Nowé, 2014).

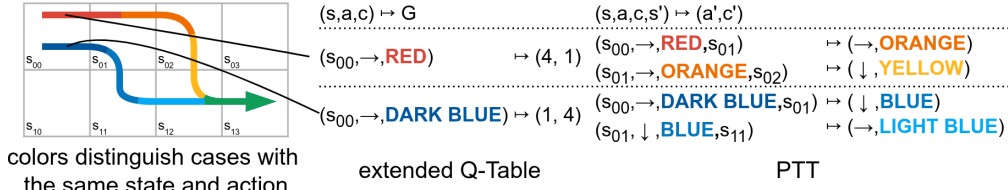

Figure 2: **Colors distinguish trajectory transitions;** examples of the extended Q-Table and Policy Transition Table (PTT) illustrate their mapping to returns and policy following.

Our method follows a policy tracking approach similar to Multi-Pareto Q-learning (Ruiz-Montiel et al., 2017), where we maintain multiple non-dominated trajectories and update their returns via TD learning. Building on this foundation, we formalize tracking with a trajectory-centric approach that integrates a color/label mechanism in the Q-Table and introduces a *Policy-Transition Table* (PTT) to explicitly and intuitively track policies.

The extended Q-Table $Q(s, a, c)$ maps state $s$, action $a$, and a *color* $c$[1] to a return $G$ — e.g., $(s_{00}, \rightarrow, RED)$ maps to $(4, 1)$ in Figure 2 — and the PTT records which transitions belong to the same trajectory — e.g., $(s_{00}, \rightarrow, RED, s_{01})$ maps to $(\rightarrow, ORANGE)$, and $(s_{01}, \rightarrow, ORANGE, s_{02})$ maps to $(\downarrow, YELLOW)$ — providing a clear structure for executing policies in complex multi-policy settings. In Figure 2, for the same state-action $(s_{00}, \rightarrow)$, the colors $RED$ and $DARK BLUE$ distinguish how to follow each trajectory. With the trajectory-centric approach, the agent is trained under a standard exploratory policy (e.g., $\epsilon$-greedy) and updates value estimates via TD learning, which adds new transitions to the trajectory's start, extending it backward. The algorithm is provided in Appendix A.

---

[1]a unique identifier we refer to as color for visual intuition

Once training is complete, policy execution proceeds as follows. The *user selects a policy* based on preferred trade-offs. The initial action $a$ and color $c$ follow Equation 1, where the *selection operator* $\Theta$ represents the user's choice, and the *non-dominated operator* $ND$ filters out dominated returns of the extended Q-Table; $A$ is the action set and $\mathcal{T}$ is the set of trajectories. To *execute the selected policy*, we use the PTT: after transitioning to next state $s'$ (Equation 2), we query the table with $(s, a, c, s')$ to obtain the next action $a'$ and color $c'$ (Equation 3). Alternating between Equations 2 and 3 keeps the agent on the intended trajectory $\tau$.

$$(\text{Policy Selection}) \quad (a_0, c_0) \sim \Theta\left(ND \bigcup_{a \in A;\, c\,from\,\delta_\tau^{(l,c)} \in \mathcal{T}(s_0, a)} Q(s_0, a, c)\right) \tag{1}$$

$$(\text{Transition}) \quad\quad\quad\quad\quad\quad s' \sim P(s' \mid s, a) \tag{2}$$

$$(\text{Policy Following}) \quad\quad\quad\quad (a', c') \sim \pi(s, a, c, s') \tag{3}$$

### 3.2 LEARNING STATIONARY AND NON-STATIONARY POLICIES

A stationary policy selects actions from a fixed distribution for each state (i.e., the action does not vary with time: $\pi_t(a|s) = \pi_{t'}(a|s)$), whereas a non-stationary policy explicitly depends on time ($\pi_t(a|s) = \pi(a|s, t)$). Most works focus on non-stationary policies, as they can outperform stationary ones and recover the entire Pareto front rather than only its convex part (White, 1982; Roijers et al., 2013; Hayes et al., 2021). Still, stationary policies solve problems requiring predictable behavior (i.e., taking the same action in a given state), and, because they form a subset of non-stationary policies and capture only the convex region, they operate over a smaller policy space and need to recover fewer Pareto-front points. We examine the construction of both stationary and non-stationary policies and how they relate to each other.

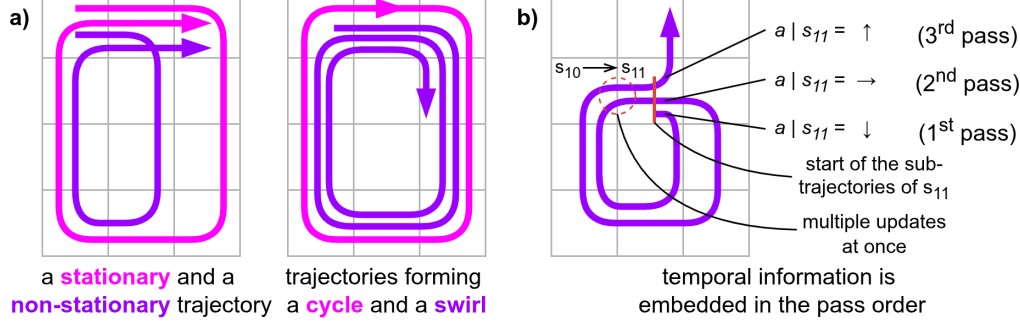

Figure 3: **Trajectory types and temporal encoding: (a)** Stationary trajectory and cycle (outer paths) vs. non-stationary trajectory and swirl (inner paths); **(b)** Time is encoded by pass order.

Under stationary policies, revisiting a state always yields the same transition, whereas non-stationary policies impose no such restriction. Thus, whether a policy is strictly stationary or generally non-stationary depends on whether we allow trajectories to form cycles (Figure 3a, left). Forming cycles produces closed, repeating behavior, ensuring strict stationarity (outer, pink trajectories), and therefore preventing cycles allows the broader set of non-stationary policies (inner, purple trajectories).

To enforce this distinction, stationary policies assign the same color to transitions as earlier transitions are added, forming a converging-trajectory tree with that color, whereas non-stationary policies, as shown in Figure 2, assign a distinct color to each transition (Equation 4, refined from Equation 3). A key consequence of assigning the same color is that stationary trajectories can block each other's expansions. For example, if the trajectories in Figure 2 were represented by the stationary coloring rule, all transitions would be *GREEN*, creating a conflict in $(s_{00}, \rightarrow)$ regarding which trajectory to follow. In practice, one of the trajectories would not have expanded to $(s_{00})$ — this issue is addressed in Section 3.5. Cycles are thus possible by connecting matching-colored transitions. Non-stationary policies, on the other hand, produce *swirls*, representing trajectories that

revisit the same states yet can take distinct paths. This representation makes it possible for swirls to emulate cycles without committing to strict stationarity (Figure 3a, right).

$$\text{(Policy Following)} \quad (a', c') \sim \pi(s, a, c, s') \quad , \begin{cases} c = c' & \text{, for stationary policies} \\ c \neq c' & \text{, for non-stationary policies} \end{cases} \tag{4}$$

Furthermore, in our trajectory-based formulation, there is no need to track time or maintain a separate history of states — time is implicit in the structure of the trajectory itself. For instance, in Figure 3b, an agent visiting state $s_{11}$ three times ($t=0, 6, 14$) the position of each occurrence in the swirl trajectory (first, second, third) determines the appropriate action (down, right, and up, respectively), allowing the agent to take different actions at each occurrence of $s$ without referencing an explicit time step $t$. This implicit encoding offers a key advantage: conditioning explicitly on time $t$, $\pi(a|s, t)$, or history $h$, $\pi(a|s, h)$, requires that the same state must be revisited at multiple distinct time steps (t=1,2...) or under different histories, while in our case, the agent can update the return estimates for all occurrences of a state in all trajectories at once, leading to a greater sample efficiency (e.g., when transitioning from $s_{10}$ to $s_{11}$, in Figure 3b) Moreover, each transition in a given trajectory also marks the start of a well-defined sub-trajectory, capturing the information from that point forward. In Figure 3b, state $s_{11}$, there are three sub-trajectories: one covering all three visits, one for the last two, and one with the final visit (i.e., the different tail sizes of the trajectory).

### 3.3 SPURIOUS DOMINATION IN CYCLES, SWIRLS, AND REGULAR TRAJECTORIES

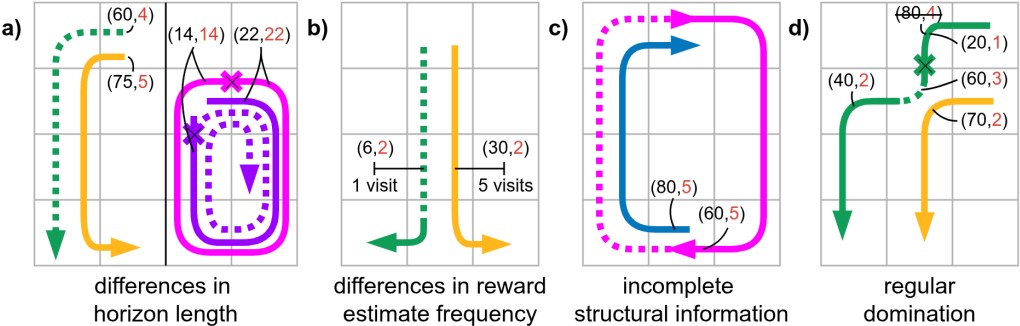

Figure 4: **(Spurious) Dominations.** Dotted lines: dominated paths; 'x': early termination; tuples: return and length (simplified, length in red). Spurious domination due to **(a)** differences in length of trajectories (left), and cycles/swirls (right); **(b)** differences in reward estimate frequency; **(c)** undetected cycle (outer path). **(d)** Regular domination: dominated trajectory gets split and later updated.

*Domination* occurs when a trajectory $\tau_1$ yields a higher return than $\tau_2$. *Spurious* (false) dominations arise when there is: (i) *differences in horizon length*: when a longer trajectory may seem better simply by having more opportunity to accumulate reward, even if a shorter one could eventually yield a higher return (Figure 4a, left). This problem is exacerbated with cycles or swirls, which repeatedly dominate the very transitions that support them (Figure 4a, right); (ii) *reward estimate frequency* (reward-frequency): frequently visited transitions converge faster to stable reward estimates, tending to spuriously dominate less visited ones (Figure 4b); or (iii) *incomplete structural information*: when undetected cycles may be incorrectly dominated by higher-return trajectories, despite their structural ability to yield arbitrarily longer paths and greater returns (Figure 4c).

To address (i), we include a step-based reward component encoding *trajectory length* $l$ (Equation 5). In strictly positive-reward settings, a negative length term prevents trivial domination by penalizing unnecessarily long trajectories; in strictly negative-reward settings, a positive term allows the agent to learn longer trajectories that would otherwise be dominated due to compounding penalties; while in mixed rewards, both cases may occur. We handle all cases uniformly by treating length as a negative reward and applying a min-shift to all rewards. This normalization, along with the uniform growth of the trajectory length, supports fair comparisons and consistent treatment of trajectory length. To address (ii), we use *average rewards* (Equation 6), normalizing cumulative rewards $r_{sum}$ by visit count $r_{count}$ for balanced convergence. Together, (i) and (ii) address spurious

domination in non-stationary policies, where trajectories and swirls extend naturally. We address (iii) in Section 3.6, where cycle information is explicitly managed.

*(Trajectory Length)*
$$l(s, a, c) = \begin{cases} l(s', a', c') + 1 & \text{, if } (s', a', c') \in \tau \\ 1 & \text{, if } (s', a', c') \notin \tau \end{cases} \tag{5}$$

*(Average Reward)*
$$\bar{r}(s, a) = r_{sum}(s, a) / r_{count}(s, a) \tag{6}$$

Ultimately, domination arises only when a shorter trajectory $\tau_s$ yields higher returns than an existing trajectory $\tau$. When domination occurs, $\tau$ is split into a valid sub-trajectory from its tail and a dangling, fragmented path of transitions (its start, or simply head), which is now outdated and must be re-learned — as in the green trajectory, Figure 4d (this issue is mitigated in Section 3.5). If left outdated, the agent may eventually reach a state missing the corresponding PTT entry (from Section 3.1) even though the trajectory is unfinished from the agent's perspective (i.e., the remaining trajectory length does not match the end). Once learning is complete, the length component can be omitted during policy selection, leaving only the trajectories that maximize the original task rewards.

## 3.4 INFINITE HORIZON BEHAVIOR AND UNDISCOUNTED RETURNS

With the inclusion of positive cycles and swirls, the problem naturally shifts from an episodic to an infinite horizon setting, where cumulative returns may grow unbounded. Although we define trajectories with well-defined return $G$ and length $l$, the issue remains as the trajectories can still grow unbounded in length. A common solution is to apply discounting to ensure returns converge. However, this undermines interpretability: the final return no longer reflects the total reward collected, but rather a weighted sum that is harder to interpret. Additionally, discounting the trajectory length component has the same effect as bounding the trajectories by a *maximum length*, which raises the question of whether finite trajectories can approximate infinite-horizon behavior.

Therefore, a policy can be defined not as a single trajectory but as a sequence of length-bounded trajectories, allowing the agent to run indefinitely with the agent adjusting collected returns (across subsequent trajectories) to preserve the chosen *average return* (Equation 7). This modeling also allows swirls to emulate cycles: once a trajectory ends, the agent can repeatedly reuse overlapping parts of the swirl. Infinite trajectories are thus unnecessary, and maximum length can be tuned, in particular when longer trajectories add no further gain in maintaining the average return.

*(Average Return)*
$$\bar{G}(s, a, c) = \bar{G}(s, a, c) + \frac{\bar{r}(s, a) - \bar{G}(s', a', c')}{l(s', a', c') + 1} \tag{7}$$

With the average return defined, we introduce the Q-function (Equation 8), capturing the return $G$ as the product of the average return $\bar{G}$ and trajectory length $l$. Together, the components introduced so far suffice for non-stationary policies; the following sections continue to address stationary ones.

*(Q-Function)*
$$Q(s, a, c) = G(s, a, c) = \bar{G}(s, a, c) \times l(s, a, c) \tag{8}$$

## 3.5 IDENTIFYING AND MANAGING STATIONARY TRAJECTORIES

As noted in Section 3.2, stationary trajectories may block each other, since the same state-action pair cannot lead to multiple trajectories with the same color. In Figure 5a (left), two trajectories diverge and later converge. Only one can retain the shared color, forcing the other to stop or be recolored. Recoloring, however, breaks the stationarity guarantee in Equation 4, even though the trajectories remain stationary, which can cause the agent to miss many stationary trajectories.

We propose (i) decomposing each trajectory into *segments* $\sigma$ — sequences of transitions that forms a sub-partition of trajectories — and assigning a unique color to each segment, $\sigma^{(c)}$, so that whenever a trajectory branches into a new path, a new color is assigned to the forming segment; (ii) introducing a *Stationary Segments Mapping* (SSM) that tracks such colored segments by their position in the trajectory (indexed by the remaining length) — depicted in Figure 5a (right). To preserve stationarity across trajectories, the agent must track the order of visited segments; and (iii) applying cycle detection (discussed in Section 3.6) to prevent trajectories from extending past a cycle and

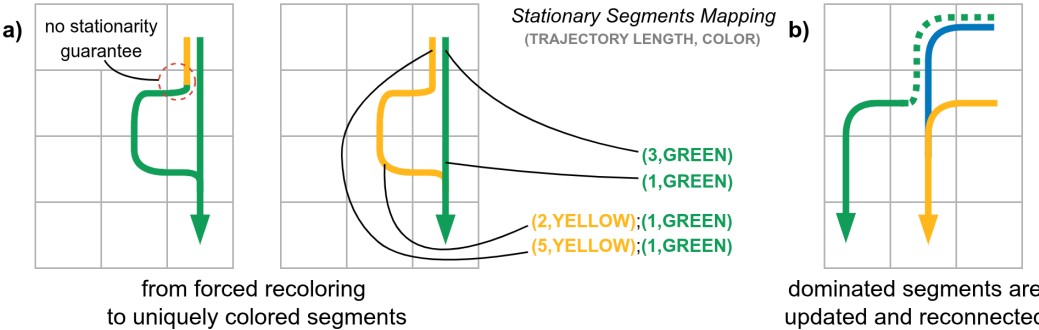

Figure 5: **Managing stationary trajectories.** Dotted lines show dominated paths. **(a, left)** Stationary trajectory must be recolored; **(a, right)** We propose uniquely-colored segments and a Stationary Segments Mapping. **(b)** Revisited domination: the segment is updated and reconnected (in blue).

becoming non-stationary. This approach replaces the way we color stationary policies, transforming the colored trajectory trees into colored-segments trees. Each new branching introduces a new color, allowing stationary trajectories to avoid blocking one another and preserve stationarity. The policy-following rule is accordingly updated from Equation 4 to Equation 9:

$$\textit{(Policy Following)} \quad (a', c') \sim \pi(s, a, c, s'), \begin{cases} c = c' \text{ or } c \neq c' & \text{for stationary policies (w/ SSM)} \\ c \neq c' & \text{for non-stationary policies} \end{cases} \quad (9)$$

To preserve the structure of these segment trees, no two segments may share the same color. This requires revisiting trajectory domination (Section 3.3). When a domination occurs, the split head segment now receives a new color $c$, updated return $G$ and length $l$, and is reconnected to the dominant trajectory (Figure 5b). In addition to the trajectory return $G$ and length $l$, we also store the segment-level return $G$ and length $l$, which under these domination dynamics, will remain up-to-date. If a length mismatch is detected when switching segments, the entire segment can be updated without revisiting every state (as observed in Section 3.3), improving sample efficiency.

## 3.6 PERFORMING CYCLE DETECTION

Cycles offer a clear benefit of representing an infinite horizon with a well-defined return while remaining bounded in length. However, detecting cycles is nontrivial due to an inherent ambiguity. As illustrated in Figure 6a, when connecting transitions from $s_{10}$ to $s_{11}$, in faded green, it is unclear whether this connection forms a cycle (*ORANGE*) — that is, whether the trajectory eventually leads back to itself — or merely encounters a separate trajectory and forms a longer path (*BLUE*), since both cases yield the same return $G$ and length $l$. In addition to the ambiguity, cycles often lack a well-defined end state (Section 3.3), causing them to dominate their tail transitions and rendering incoming trajectories repeatedly outdated. If a maximum length is imposed for the trajectories, the cycle eventually exhausts that maximum, preventing any other state from reaching the cycle.

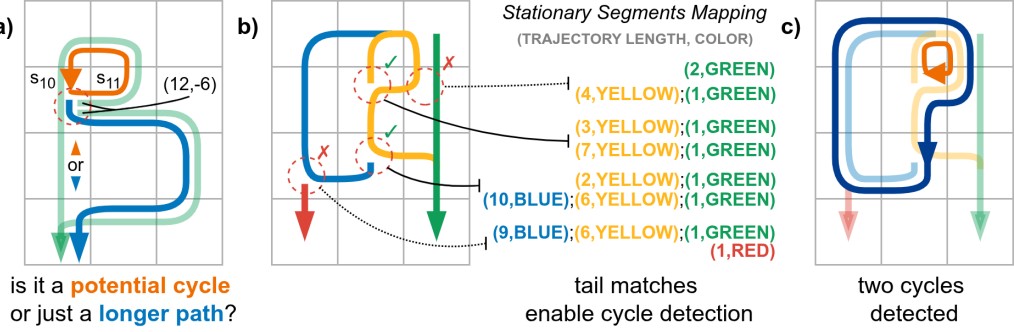

Figure 6: **Performing cycle detection.** Tuples denote return and length (simplified). **(a)** Ambiguity: cycle or longer path? **(b)** Tail matches confirm a cycle. **(c)** Resulting detected cycles.

Building on the colored-segments trees defined in Section 3.5, we propose the following cycle detection mechanism. If the potential connection occurs between transitions with the same color $c$ (e.g., *YELLOW* to *YELLOW*, Figure 6b), they must belong to the same segment, and a cycle can be safely formed. If the colors differ, we must determine whether the transitions lie on the same trajectory by checking whether the shorter trajectory is a tail (or suffix) of the longer one. To verify this, we use the SSM (Section 3.5) to compare the segments by color $c$ and length $l$. If one is indeed a suffix of the other, the transitions belong to the same trajectory (e.g., *BLUE* to *YELLOW*); otherwise, the paths have diverged and represent distinct trajectories (e.g., *RED* to *BLUE* and *YELLOW* to *GREEN*), as shown in Figure 6b by the red *Xs*.

Once a cycle is detected (two resulting cycles in Figure 6c), we store a copy of its transitions for two reasons. First, all cycle transitions receive a new color $c$, with each transition marked as part of the cycle and sharing the same return $G$ and length $l$. This copy ensures the original unmodified trajectories remain stationary. With cycle detection in place, the agent avoids exhausting maximum trajectory length budgets and repeatedly dominating its own transitions (discussed in Section 3.3).

Table 1: **Ablation plan.** Each row tests a design choice with metrics and expected outcomes.

S - Stationary; NS - Non-stationary; NPS - Naive Policy Selection; PF - Pareto front; ND - non dominated.

| Id | Ablation | Hypothesis / Experiment / Metrics / Results |
|---|---|---|
| AS1 | Policy tracking *vs.* state-wise ND selection | Lack of trajectory-level tracking causes agents to follow dominated policies. **Experiment:** NPS (i.e., random non-dominated actions) vs. S and NS policies. **Metrics:** Policy coverage. **Results:** NPS gets less treasure value because it combines state-wise ND actions into trajectories that often yield dominated solutions (Figure 7a). |
| AS2 | Learning the full/convex PF with NS/S policies | Allowing NS policies enables full PF recovery; restricting to S policies limits recovery to its convex subset (thus lower hypervolume). **Experiment:** S vs. NS policies. **Metrics:** Policy coverage; convergence speed. **Results:** NS polices recover the full PF, whereas S ones recover only the convex subset (Figure 7a and 7b). |
| AS3 | Horizon-length bias in domination | Longer trajectories spuriously dominate others without normalization. **Experiment:** Disable trajectory length. **Metrics:** Hypervolume over time. **Results:** No observable difference, as the environment already incorporates a length-like component, time penalty (Figure 7b). |
| AS4 | Reward-frequency bias in domination | Reward frequency biases cause frequent low-reward policies to appear superior to sparse high-reward ones. **Experiment:** Disable average reward. **Metrics:** Hypervolume over time. **Results:** Imbalanced reward frequencies destabilize learning and degrade performance (Figure 7b). |
| AS5 | Structural-info gaps in domination | Failure to detect cycles biases Pareto comparisons, making cycles appear dominated by high-return trajectories. **Experiment:** S without cycle detection. **Metrics:** Hypervolume over time. **Results:** Failing to detect cycles sharply degrades performance (Figure 7b). |
| AS6 | Maximum trajectory-length sensitivity and policy selection | Maximum trajectory length affects whether policies sustain stable returns. **Experiment:** Vary max length (2, 5, 10, 20, 30). **Metrics:** Hypervolume from promised (max_length steps) vs. realized (1,000 steps). **Results:** Promised and realized returns align, indicating sustained return over following multiple trajectories (Figure 7c). |
| AS7 | Blocking segments and SSM | SSM prevents blocking and coloring conflicts, enabling stationary policies to be recovered correctly. **Experiment:** Disable SSM. **Metrics:** Hypervolume from promised (max_length steps) vs realized (1,000 steps). **Results:** Promised and realized returns not only align but also improve, confirming correct recovery with SSM (Figure 7c). |
| AS8 | Split-and-reconnect segment updates | Applying segment updates improves sample efficiency and accelerates recovery after dominated trajectories. **Experiment:** Disable reconnection. **Metrics:** # of splits; # of mismatches. **Results:** Split-and-reconnect yields substantially fewer trajectory-length mismatches (Figure 7d). |

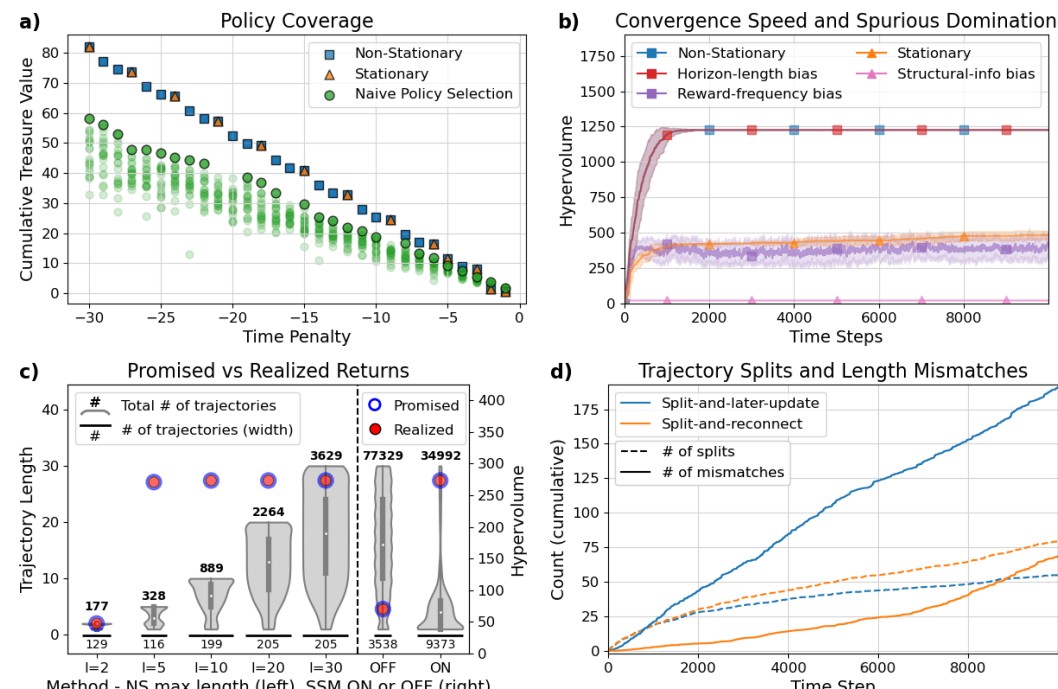

Figure 7: **Ablation study results.** Each plot shows different aspects of policy evaluation: (a) returns for Stationary, Non-Stationary, and naive policies; (b) convergence and spurious domination; (c) trajectory lengths and promised vs realized returns; (d) trajectory splitting and length mismatches.

## 4 ABLATION STUDIES

Existing tabular algorithms (Van Moffaert & Nowé, 2014; Van Moffaert & Nowé, 2017) do not address the infinite-horizon setting, and because our methods are best viewed as a conceptual continuation of Ruiz-Montiel et al. (2017), a direct comparison is not meaningful. We therefore validate our algorithmic contributions through targeted ablations.

We analyze key design choices through ablations on the well-known DeepSeaTreasure (DST) (Vamplew et al., 2011) problem in MO-Gymnasium (Alegre et al., 2022). DST is a gridworld that tests the trade-off between treasure value and time (i.e., choosing between shallow, low-value treasures and deeper, higher-value ones). To adapt DST to an infinite-horizon setting, we make terminal states non-terminal and allow the collection of the same treasure multiple times. For simplicity and to remain close to the original task, we teleport the agent via an $\epsilon$-transition with a reward of $(0, 0)$, thereby achieving the intended behavior. Agents are trained for 10,000 steps, averaging results over 20 seeds. Each *ablation study* (AS1-AS8) evaluates a specific aspect from Section 3, with Figures 7a–7d presenting the main results, and hypothesis, experiments, metrics, and results summarized in Table 1, and further ablation explanations are below the table.

**Figure 7a (AS1, AS2)** reports returns for non-stationary (NS), stationary (S), and naive policy selection (NPS, i.e., selecting random non-dominated actions). We tested whether policies achieved the learned returns on average, with NPS using the same target returns as NS. NS policies recover the full Pareto front, whereas S policies cover only the convex subset, and NPS yields many dominated solutions. Note that NPS only performs better for small time penalties ($-1$ and $-2$) because it captures better treasure/time penalty on average, while NS and S obtain their learned return.

**Figure 7b (AS2–AS5)** shows hypervolume[2] convergence for S and NS policies, around 1,000 time steps. The figure also highlights spurious domination, represented by 3 biases: horizon length bias (red), reward-frequency bias (purple), and structural information (pink). Note that square markers are all for NS and triangles for S policies. Disabling trajectory-length (AS3) does not affect horizon-

---

[2]Hypervolume measures the objective space dominated by obtained solutions relative to a reference point. Larger values indicate better *convergence* (solutions near the true PF) and *diversity* (well-spread along the PF).

length bias (red line), as the time penalty already acts as a trajectory length, balancing the returns. In contrast, the purple line shows that the reward-frequency bias (AS4) case causes slight instability and lower performance. Disabling cycle detection in S (AS5) removes structural guidance, further limiting performance (pink line with almost no hypervolume).

**Figure 7c (AS6, AS7)** examines maximum-length sensitivity in NS policies and the effect of enabling SSM in S policies. Violin plots show trajectory length distributions, with *promised* returns representing values presented to the user and *realized* returns representing values obtained after 1,000 steps. The match between promised and realized returns (right Y axis shows the hypervolume) indicates the agent sustains average returns over longer horizons. In DeepSeaTreasure, the return $(8.2, -3)$ yields the best average among all treasures, explaining why $l = 2$ underperforms, whereas $l \geq 5$ achieves greater hypervolume. Enabling SSM is crucial for stationary policies: without it, cycle detection is disabled, limiting long-horizon performance. The figure also shows that SSM causes trajectories to be shorter, as cycle detection halts growth once a cycle forms.

**Figure 7d (AS8)** reports splits (dashed lines) and mismatches (solid lines). Mismatches occur when trajectories end prematurely, while splits indicate domination. Since splits are similar, lower mismatches for Split-and-Reconnect (orange lines) demonstrate the agent quickly reconnects and updates segments, leading to a higher sample efficiency.

## 5 EXTENDING DESIGN PRINCIPLES TO DEEP RL

While we identified general issues and solutions in extending multi-policy methods from episodic to infinite-horizon settings, the mechanisms presented here focus on tabular methods, which are exact and discrete. We now outline how these design principles might be extended to deep RL, where solutions are approximate and continuous — an area that remains largely open.

One possible direction is to recast PTT and Q-table within an actor–critic framework. The actor models the policy $\pi(a, c'|s, c)$, where the color $c$ becomes a learned embedding; for stationary policies, this embedding effectively identifies the policy (i.e., a policy embedding). The critic, in turn, models $Q(s, a, c)$ in the same way that standard deep RL methods model $Q(s, a)$.

The SSM structure could be implemented as a Graph Neural Network (GNN) (Wu et al., 2019), where each colored segment corresponds to a node, and the connections between segments become the edges. Although this graph is smaller than encoding all transitions of all policies, it remains large, which could strain the GNN capacity; in particular, reliably capturing cyclic structure is a known challenge for current GNN architectures.

For non-stationary policies, another direction is to drop color embedding entirely and adopt a return-conditioned actor, similar to PCN Reymond et al. (2022), while designing the critic to model non-dominated returns for each state–action pair $(s, a)$. One possible approach is to use density estimation with deep generative models (Bond-Taylor et al., 2021), allowing the critic to capture the returns as a distribution and then approximate this distribution to recover the Pareto front.

## 6 CONCLUSION

We introduced design principles that extend TD-based multi-policy MORL from episodic to infinite-horizon settings. Our trajectory-centric framework enables reliable tracking and execution of both stationary and non-stationary policies, supports cycle detection, and prevents spurious domination from horizon mismatches or incomplete information. Through ablation studies, we showed how each principle contributes to recovering diverse, interpretable, and stable policies. Looking ahead, these principles provide a foundation for extending TD-based methods to deep settings, where scalable representation and online adaptation remain open challenges. By ensuring well-defined behavior in infinite horizons, our framework provides a path toward practical multi-policy MORL systems that can support preference-driven decision-making in complex domains.

### REPRODUCIBILITY STATEMENT

We have taken several measures to ensure reproducibility. A detailed description of our algorithm, training procedures, and evaluation protocols is provided in Section 3 and 4 of the main text, with

further implementation details in Appendix A. We also provide our code and scripts in the supplementary materials, which enable end-to-end reproduction of all experiments.

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

# A   ALGORITHMIC DETAILS AND ANALYSIS

Tabular RL in the infinite-horizon setting follows the structure in Algorithm 1. The algorithm maintains a Q-Table of state–action values, updated incrementally as the agent interacts with the environment. At each step, the agent selects an action via an exploration policy (e.g., $\epsilon$-greedy), observes the reward and next state, and applies the TD update on the Q-Table. As usual, the discount factor $\gamma$ weights future rewards, while the learning rate $\alpha$ controls how quickly new information overrides old estimates. Repeated interactions refine the Q-values, which can then define a greedy policy. In Sections A.1 and A.2, the *Q-Table update* is replaced by our non-stationary and stationary *Multi-Policy Update*, respectively, and Section A.3 discusses their time and space complexity.

---

**Algorithm 1** Tabular RL for Infinite-Horizon

---

**Require:** State space $\mathcal{S}$, action space $\mathcal{A}$, learning rate $\alpha$, discount factor $\gamma$, exploration policy $\pi_\epsilon$, maximum steps $T$
1:  Initialize $Q(s, a)$ arbitrarily for all $s \in \mathcal{S}, a \in \mathcal{A}$
2:  **for** $t = 1$ **to** $T$ **do**
3:      Observe current state $s_t$
4:      Select action $a_t$ according to $\pi_\epsilon(Q, s_t)$
5:      Execute $a_t$, observe reward $r_t$ and next state $s_{t+1}$
6:      **Update Q-Table:**

$$Q(s_t, a_t) \leftarrow Q(s_t, a_t) + \alpha \Big[ r_t + \gamma \max_{a'} Q(s_{t+1}, a') - Q(s_t, a_t) \Big]$$

7:      $s_t \leftarrow s_{t+1}$
8:  **end for**
9:  **return** $Q(s, a)$

---

## A.1   NON-STATIONARY

Algorithm 2 outlines the update procedure for the non-stationary multi-policy. The method extends standard TD updates to maintain multiple policies simultaneously via the Q-Table and a Policy-Transition Table — PTT (from Section 3.1).

**Step 1.** Retrieve from Q-Table the non-dominated set at the next state $s'$, consisting of tuples $(s', a', c')$ whose Q-values are non-dominated.

**Step 2.** If the set is empty, then $s'$ is being visited for the first time and $Q(s', a', c')$ has no entries. In this case, add a new Q-value estimate for $(s, a, c)$ from reward $r$ with trajectory length 1.

**Step 3.** Fetch all PTT entries $(s, a, c, s', a', c')$ associated with the current transition $(s, a, s')$. Delete PTT entries whose successors $(s', a', c')$ are no longer in the non-dominated set (obsolete). Their Q-value estimates (i.e., $(s, a, c)$ restart with reward $r$ and trajectory length 1. Add any entries from the non-dominated set to the PPT, associating them with the current $(s, a)$ and generating a new color $c$ for each association $(s, a, c, s', a', c')$, thereby extending trajectories from $s'$ to $s$ with distinct colors $c_i$ and adding 1 to the trajectory length.

**Step 4.** Use each updated PTT entry to update the Q-Table according to Equations 7 and 8.

**Step 5.** Prune dominated entries from $(s, a)$ in the Q-Table and then in the PTT, preserving only Pareto-optimal $(s, a, c)$ entries for future updates.

## A.2   STATIONARY

Algorithm 3 outlines the update procedure for the stationary multi-policy. In addition to the Q-Table and Policy Transition Table (PTT), it incorporates the Stationary Segments Mapping — SSM (from Section 3.5).

**Step 1.** Retrieve from Q-Table the non-dominated set at the next state $s'$, consisting of tuples $(s', a', c')$ whose Q-values are non-dominated.

---

**Algorithm 2** Multi-Policy Update: Non-Stationary

---

**Require:** Current state $s$, action $a$, next state $s'$, Q-Table, Policy Transition Table (PTT)
 1: **Step 1: Fetch ND entries**
 2: next_state_nd_set ← fetch_nd_set($Q, s'$)          ▷ $(s', a', c')$ entries
 3: **Step 2: Initialize Q-Table if no ND entries at s'**
 4: **if** next_state_nd_set is empty **then**
 5:     $c$ ← generate_new_color()
 6:     *ADD entry $(s, a, c)$ to Q-Table*
 7: **end if**
 8: **Step 3: Fetch PTT entries for transition (s, a, s'), add new and discard old transitions**
 9:    ptt_entry_set ← fetch_ptt_entries($PTT, s, a, s'$)      ▷ $(s, a, c, s', a', c')$ entries
10:    ptt_old_entries ← ptt_entries_set − next_state_nd_set    ▷ Compare $(s', a', c')$, keep PTT entry
11: **for** $(s, a, c, s', a', c')$ in ptt_old_entries **do**
12:     *DELETE entry $(s, a, c, s', a', c')$ from PPT*
13: **end for**$(s, a, c, s', a', c')$
14:    ptt_new_entries ← next_state_nd_set − ptt_entries_set     ▷ Compare $(s', a', c')$, keep $(s', a', c')$
15: **for** $(s', a', c')$ in ptt_new_entries **do**
16:     $c$ ← generate_new_color()
17:     *ADD entry $(s, a, c, s', a', c')$ to PTT*
18: **end for**
19: **Step 4: Update Q-Table for each PTT entry**
20: **for** $(s, a, c, s', a', c')$ in ptt_entries **do**
21:     *ADD/UPDATE entry (s, a, c) to Q-Table*        ▷ Update done via Equation 7 and 8
22: **end for**
23: **Step 5: Clean up dominated entries**
24: *state_action_dominated_set ← fetch_dominated_set(Q, s, a)*
25: **for** $(s, a, c)$ in state_action_dominated_set **do**
26:     *DELETE entry $(s, a, c)$ from Q-Table*
27:     *DELETE entry from PTT which starts with $(s, a, c)$ as in $(s, a, c, s', a', c')$*
28: **end for**

---

**Step 2.** If the set is empty, then $s'$ is being visited for the first time and $Q(s', a', c')$ has no entries. In this case, add a new Q-value estimate for $(s, a, c)$ from reward $r$ with trajectory length 1.

**Step 3.** Fetch all PTT entries $(s, a, c, s', a', c')$ associated with the current transition $(s, a, s')$. **Unlike the non-stationary case,** no entries are obsolete: dominated trajectories are reconnected. Add non-dominated successors from Step 1 to the PTT, linking them to $(s, a)$ with the appropriate color $c$. Assign a new color if $(s, a, c)$ is already reached by another entry; otherwise, reuse the existing color, thereby extending trajectories from $s'$ to $s$ with trajectory length $+1$.

**Step 4.** Update the Q-Table using each revised PTT entry, according to Equations 7 and 8.

**Step 5.** Check for cycles. If one is found, duplicate its transitions and assign them a new shared color; otherwise, update the SSM with the new PTT entry.

**Step 6.** Prune dominated entries from $(s, a)$ in the Q-Table. Reconnect all prior associations from pruned entries to their dominant counterparts in the PTT, updating their Q-Values, and generating new colors as in Step 3. This preserves only Pareto-optimal $(s, a, c)$ entries for future updates.

## A.3 Algorithmic Complexity

We summarize the worst-case computational complexity of the stationary and non-stationary multi-policy updates.

The **non-stationary update** is dominated by computing the non-dominated set for the next state (Step 1 in Algorithm 2). Let $M$ denote the number of objectives and $N_{s'}^{ns}$ the total number of Q-value estimates for all actions $a'$ in the next state $s'$, found in *non-stationary policies*. The worst-case complexity is $O(MN_{s'}^{ns\,2}) \sim O(N_{s'}^{ns\,2})$, as the number of objectives is constant. This arises from pairwise comparisons of all Q-value vectors in $s'$ across objectives.

---

**Algorithm 3** Multi-Policy Update: Stationary

---

**Require:** Current state $s$, action $a$, next state $s'$, Q-Table, Policy Transition Table (PTT), Stationary Segments Mapping (SSM)

1: **Step 1: Fetch ND entries**
2: next_state_nd_set ← fetch_nd_set($Q, s'$)        ▷ $(s', a', c')$ entries
3: **Step 2: Initialize Q-Table if no ND entries at s'**
4: **if** next_state_nd_set is empty **then**
5:   $c$ ← generate_new_color()
6:   *ADD entry $(s, a, c)$ to Q-Table*
7: **end if**
8: **Step 3: Fetch PTT entries for transition (s, a, s') and add new transitions**
9: ptt_entry_set ← fetch_ptt_entries($PTT, s, a, s'$)     ▷ $(s, a, c, s', a', c')$ entries
10: ptt_new_entries ← next_state_nd_set − ptt_entries_set   ▷ Compare $(s', a', c')$, keep $(s', a', c')$
11: **for** $(s', a', c')$ in ptt_new_entries **do**
12:   $c$ ← fetch_or_generate_color(s', a', c')   ▷ Repeat color if first extension, new one otherwise
13:   *ADD entry $(s, a, c, s', a', c')$ to PTT*
14: **end for**
15: **Step 4: Update Q-Table for each PTT entry**
16: **for** $(s, a, c, s', a', c')$ in ptt_entries **do**
17:   *ADD/UPDATE entry (s, a, c) to Q-Table*       ▷ Update done via Equation 7 and 8
18: **end for**
19: **Step 5: Check for cycles and update the SSM**
20: **for** $(s, a, c, s', a', c')$ in ptt_entries **do**
21:   **if** cycle is detected in PTT **then**
22:    *Identify all transitions in the cycle*
23:    $c$ ← generate_new_color()
24:    **for** each transition in the cycle **do**
25:     *DUPLICATE the transition*
26:     *UPDATE SSM with the new PTT entry*
27:    **end for**
28:   **else**
29:    UPDATE SSM with the new PTT entry
30:   **end if**
31: **end for**
32: **Step 6: Clean up dominated entries**
33: dominated_set ← fetch_dominated_set(Q, s, a)
34: **for** each d:$(s, a, c)$ in dominated_set **do**
35:   **for** each PTT entry ending in $(s, a, c)$, i.e., $(s_{prev}^d, a_{prev}^d, c_{prev}^d, s, a, c)$ **do**
36:    $c_{\text{new}}$ ← fetch_or_generate_color()        ▷ As in Step 3
37:    *RECONNECT previous links with color c and update Q-Table and PTT with $c_{new}$*
38:   **end for**
39:   *DELETE Q-Table entry $(s, a, c)$*
40:   *DELETE PTT entries starting from $(s, a, c)$, i.e., $(s, a, c, s', a', c')$*
41: **end for**

---

In addition to doing the same work as the non-stationary update, the **stationary update** also iterates over stationary segments (from SSM) when checking for cycles (Step 5 in Algorithm 3) and over outdated segments to update and reconnect them (Step 6 in Algorithm 3). Let $S$ denote the number of stationary segments, $N_{s'}^s$ the total number of Q-value estimates for all actions $a'$ in the next state $s'$, found in *stationary* policies, $N_{sa}$ the number of Q-value estimates for the current state-action pair $(s, a)$ used for cycle detection, $E_D$ and $E_d$ the numbers of dominant and dominated entries, respectively, and $Ld$ the dominated trajectory length. The worst-case complexity is $O(MN_{s'}^{s\,2} + SN_{sa}N_{s'}^s + L_dE_dE_D) \sim O(N_{s'}^{s\,2} + SN_{sa}N_{s'} + L_dE_dE_D)$, where the first term corresponds to computing the non-dominated set, the second to iterating over stationary segments for cycle detection, and the third to updating dominated segments. Here, $N_{s'}^{s\,2} \geq N_{sa}N_{s'}^s$ since $N_{sa}$ counts only the estimates for the current action. In the worst case, $S$ can be as large as the trajectory length of the smaller trajectory in each pair comparison. Since stationary policies form a

subset of non-stationary policies, we have $N_{s'}^s \leq N_{s'}^{ns}$, which can partially offset the difference in computational complexity.

For both stationary and non-stationary updates, **space complexity** is dominated by the Q-Table and PTT. The Q-Table stores one entry per non-dominated trajectory for each state-action pair, requiring at most $O(|S||A|N_{\max})$ space, where $N_{\max}$ is the maximum number of non-dominated trajectories per state-action. The PTT stores the transitions leading to these trajectories, with a worst-case bound of $O(|S|^2|A|N_{\max})$, though in practice it is much smaller. Similarly, the SSM stores stationary segments, with space proportional to the number of non-dominated segments. Because the number of stationary segments is smaller than the number of transitions, the SSM is smaller than the PTT.

Regarding the color space $|C|$, for non-stationary policies, the color only needs to be unique within each state–action pair, since the Q-table is already indexed by (s, a) and the PTT only links (s, a, c, s') to (a', c'). Thus, the space of possible colors is bounded by the maximum number of visits to each state–action pair. In contrast, for stationary policies, the color must be unique per policy. By tracking how many transitions each color participates in, we can upper-bound the number of required colors by $|S|$ (the state space), since in the worst case, each state may become the root of a tree of converging trajectories. In practice, this bound is much smaller because colors propagate along trajectories. In the best case, the bound is $|P|$ (the policy space), one color per policy.

## B  USE OF LARGE LANGUAGE MODELS (LLMS)

We used Large Language Models (LLMs) to support writing and research. Specifically, they were employed to polish and refine phrasing for clarity and conciseness, to assist in exploring alternative framings of ideas (e.g., refining how concepts and contributions are presented, not generating new research directions), to complement traditional tools (e.g., Google Scholar) in identifying related work, and to provide limited coding support, such as boilerplate code for plotting or debugging assistance. The models did not contribute at the level of a coauthor, and all research design, implementation, analysis, and conclusions are our own.

