# OpenReview forum: "Design Principles for TD-based Multi-Policy MORL in Infinite Horizons"
_ICLR.cc/2026/Conference — ICLR 2026 Conference Withdrawn Submission_

### Official Review · Reviewer_rxHq · 2025-10-29

**Soundness:** 2
**Presentation:** 1
**Contribution:** 2
**Rating:** 2
**Confidence:** 3

**Summary:**

MORL addresses multiple conflicting objectives by approximating the Pareto Front (PF)—a set of non-dominated policies representing trade-offs. The authors critique existing methods: supervised approaches such as Pareto Conditioned Networks (PCNs) require costly retraining and curated data, limiting online adaptation, while TD-based methods are confined to tabular, episodic tasks. They propose a trajectory-centric framework using colour-labelled transitions, tables for policy tracking, and mechanisms to handle stationary/non-stationary policies, spurious dominations and cycles. Ablation studies on an adapted DeepSeaTreasure environment validate the principles, demonstrating improved policy diversity and reliability. The paper concludes by positioning these principles as a foundation for deep RL extensions. Appendices detail algorithms and complexity analysis.

**Strengths:**

1.	The paper is structured as a series of "design principles," and the methodology attempts to solve each identified problem in sequence (e.g., policy tracking, spurious domination, cycle detection). This is a logical way to build a complex algorithm.
2.	The ablation study (Section 4) shows contribution of each "design principle" to the final performance (e.g., AS1 shows the need for policy tracking, AS5 shows the need for cycle detection). This provides clear empirical justification for the framework's internal components.

**Weaknesses:**

1. Clarity and Presentation: The paper introduces a dense vocabulary of new terminology (e.g., "color-label," "swirl trajectory," "Policy-Transition Table (PTT)," "Stationary Segments Mapping (SSM)") without sufficient formal definition or high-level intuition. Section 3 is exceptionally difficult to follow, making the core methodology hard to assess, reproduce, or build upon.
2. Disconnect Between Motivation and Future Extension: The paper motivates itself as laying a "principled foundation for future deep-RL extensions." However, the entire methodology is tabular, and the experiments are confined to a single 2D gridworld. The paper makes no attempt to explain how complex, discrete structures like the PTT and SSM could ever be scaled to the high-dimensional, continuous state spaces required by deep RL. This makes the primary motivation feel unsupported.
3. Insufficient Comparison to Baselines: While the internal ablation study is useful, the paper fails to compare its final algorithm against MORL baselines. Even if SOTA deep methods are unsuitable, comparisons against adapted tabular methods (e.g., Pareto Q-Learning https://jmlr.org/papers/volume15/vanmoffaert14a/vanmoffaert14a.pdf) are necessary to benchmark the algorithm's actual performance, sample efficiency, and computational cost.
3. Missing Theoretical Foundations (Markov Chains): The paper's unconventional treatment of "cycles" and "swirls" to manage infinite-horizon policies lacks rigor. A better analysis of infinite-horizon problems necessitates a connection to established Markov Chain theory (e.g., ergodicity, aperiodicity, etc.). The paper avoids this, making it unclear if the proposed cycle-detection and policy-tracking mechanisms are robust.

**Questions:**

1. About the "Color-Label" Mechanism, how is the "color-label" c formally defined, generated, and stored? Is c a discrete integer? Does the space of c grow unboundedly as new trajectories and segments are discovered?
2. The main premise is to build a foundation for deep RL. What is the explicit proposed path for scaling the PTT and SSM structures? Would this require approximating these discrete, graph-like tables with a GNN or a similar architecture? How would the "color-label" concept translate to a continuous state space?
3. The distinction between "cycles" (stationary) and "swirls" (non-stationary) is central. Is a "swirl" (Fig 2a) simply a non-stationary policy that revisits states? How does the "implicit" temporal encoding (Fig 2b) offer a concrete advantage over a standard formulation that includes a time-step t or history h as part of the state?
4. The paper states the environment (DeepSeaTreasure) was "adapted to an infinite-horizon setting (agent goes to the initial state after the end of an episode)." This sounds like an episodic task that is simply reset, not a true infinite-horizon, continuing task. Can the authors clarify this? If the task is truly episodic, it may undermine the paper's entire motivation about infinite-horizons.
5. The algorithmic complexity for stationary policies in Appendix A.3 is much higher than that of non-stationary policies, while the authors stated ““they may be faster to learn” in lines 163–164. This sounds contradictory.

---

> ### Author Response · Authors · 2025-11-23
>
> First, thank you for your thoughtful and specific questions — we appreciate your feedback.
>
> **W1 (same question from dUff and rxHq):**
>
> We have rewritten parts of the text to enhance clarity and improve the reader’s experience.
>
> **W2 & Q2 (same question from c5nB, dUff, and rxHq):**
>
> We agree. We initially planned to include a short discussion on this topic, but removed it because it required further development to move beyond speculation. In retrospect, we should have included a brief indication of these directions while clearly noting that they were preliminary.
>
> See Section 5 - Extending Design Principles to Deep RL.
>
> **W3 (same question from c5nB, Hujr, and rxHq):**
>
> Deep RL approaches are not directly comparable, as they belong to a different class of methods; accordingly, we position our work as addressing a gap in tabular approaches, with the goal that the resulting design principles could inspire Deep RL-based extensions.
>
> At the same time, existing tabular approaches do not apply to the infinite-horizon setting. The algorithms in this work are a natural yet non-trivial extension of these methods to this setting; therefore, adapting prior tabular baselines or comparing to them “as is” would not yield a meaningful comparison. Instead, we performed a series of ablations to validate the specific design choices in our framework.
>
> We’ve added clarification in Section 4, 1st paragraph.
>
> **W4:**
>
> We plan to incorporate more theoretical background on Markov Chains in future revisions.
>
>
> **Q1:**
>
> We refer to them as ‘colors’ for intuition, but each color is simply a unique identifier (a discrete integer). It is stored in the Q-table as (s, a, c) and in the PTT as (s, a, c, s', a', c') as trajectories are extended. In our implementation, we use UUIDs, but a sequential integer would work equally well.
> For non-stationary policies, the color only needs to be unique within each state–action pair, since the Q-table is already indexed by (s, a) and the PTT only links (s, a, c, s') to (a', c'). Thus, the space of possible colors is bounded by the maximum number of visits to each state–action pair.
> For stationary policies, the color must be unique per policy. By tracking how many transitions each color participates in, we can upper-bound the number of required colors by |S| (the state space), since in the worst case, each state may become the root of a tree of converging trajectories. In practice, this bound is much smaller because colors propagate along trajectories. In the best case, the bound is |P| (the policy space) — one color per policy.
> Both bounds are reasonable in practice.
> We have added that elaboration in Appendix A.3, 5th paragraph
>
> **Q3:**
>
> “Swirl” is coined to represent a visual counterpart to cycle. Stationary policies form cycles as they can only repeat the same transition when revisiting a state, and non-stationary policies form swirls as they are free to revisit the same states, without being restricted to a specific transition in each state. Therefore, swirls can repeat the same pattern as cycles, but are not limited to it.
>
> Regarding the temporal encoding question. When a policy is explicitly conditioned on time $t$, $\pi(a|s,t)$, or history $h$, $\pi(a|s,h)$, it requires that the same state must be revisited at multiple distinct time steps (t=1,2...) or under different histories to estimate returns, which becomes significantly less sample-efficient. In our case, all trajectories that use the given transition are updated at once.
>
> We clarified both points in the text (Section 3.2, 3rd and 4th paragraphs).
>
> **Q4:**
>
> To convert the episodic task into an infinite-horizon environment, we make terminal states non-terminal and allow the collection of the same treasure multiple times. Under this modification, the agent can act without termination and may accumulate any sequence of treasures, potentially revisiting and collecting the same one multiple times.
>
> For simplicity and to keep the modified version closer to the original in the level of challenge, we chose to teleport the agent (i.e., simply adding an ε-transition with reward (0,0)). It would be, in fact, equivalent for our purposes, as just letting the agent continue (no teleporting) or to enforce new restrictions, such as making the agent return to the start state or requiring the sub to surface, both of which alter the problems to a different task, with different levels of challenge.
>
> We added a clarification (Section 4, 2nd paragraph)
>
> **Q5:**
>
> The original statement refers to the size of the policy spaces. It would be faster in the sense that there are fewer policies to be learned, which also reduces the impact on algorithm complexity.
>
> To avoid confusion, we changed Section 3.2, 1st paragraph.
>
> We also clarified in Appendix A.3 3rd paragraph. that the policy space plays a crucial role in the complexity analysis, and we also distinguished the terms used in the complexity analysis to match that.

---

### Official Review · Reviewer_dUff · 2025-11-01

**Soundness:** 3
**Presentation:** 2
**Contribution:** 2
**Rating:** 4
**Confidence:** 3

**Summary:**

This paper presents a framework for temporal-difference (TD)-based multi-policy multi-objective reinforcement learning (MORL) in the infinite-horizon setting.
The authors propose a color-coded representation of policies and trajectories, introduce mechanisms for policy tracking and stationary segment mapping, and formalize several “design principles” (Sections 3.1–3.6) aimed at preventing issues such as spurious domination and non-stationary credit assignment.
The framework is implemented in a tabular DeepSea Treasure environment, and eight ablation studies (AS1–AS8) are reported to justify individual design components.

**Strengths:**

1. The notion of mapping stationary segments and detecting cycles to handle infinite horizons is conceptually novel and mathematically consistent.
2. Despite being primarily theoretical, the paper supports every design element with an ablation (AS1–AS8), providing empirical intuition for why each component matters.

**Weaknesses:**

1. The framework’s setting—colored trajectories, segments, and swirls—is unconventional and a bit under-explained.
A concise motivating example illustrating would make the paper much more approachable.

2. The framework remains fully tabular and is evaluated only on DeepSea Treasure.
Without evidence or discussion of scalability, it is difficult to assess real-world practicality.

3. The paper devotes substantial space to figures and tables, leaving limited room for interpretation.
For instance, Table 1 lists all ablations but uses many internal terms unfamiliar to newcomers; more textual reasoning or summary commentary would be preferable.

4. The Appendix follows immediately after References (page 11) without a clear break.
Moving large tables (e.g., Table 1) to the appendix and separating these sections with \section*{Appendix} would improve readability.

**Questions:**

1. Could the authors provide a small running example early in Section 3 to clarify the role of colors and how they differ from conventional policy identifiers?

2. How might the proposed tabular mechanisms (e.g., PTT, SSM) extend to function approximation or actor-critic settings?

3. In the ablation results (Section 4), are the reported improvements statistically significant over multiple seeds?

---

> ### Author Response · Authors · 2025-11-23
>
> First, thank you for your thoughtful and specific questions — we appreciate your feedback.
>
> **W1 (same question from dUff and rxHq):**
>
> We have rewritten parts of the text to enhance clarity and improve the reader’s experience.
>
> **W2 (same question from c5nB, Hujr, and dUff):**
>
> Classical single-objective tabular methods already struggle to scale to real-world problems, and the multi-policy setting exacerbates this challenge. We discuss algorithmic complexity in the appendix: for non-stationary policies, complexity is dominated by pairwise comparisons to filter non-dominated returns, while for stationary policies, it is limited by cycle detection.
>
> **W3:**
>
> We devoted substantial space to the methodology to fully develop our ideas. The figures were included to make these ideas more accessible, providing a visual anchor so readers can quickly connect them to the text. In particular, Table 1 condenses and organizes all proposed ablation studies, aligning with the subsections and figure order, while the main discussion is highlighted in textual form.
>
> **W4:**
>
> Fixed.
>
>
> **Q1:**
>
> The color is a unique identifier we refer to as color for visual intuition. We added an image, an example, and clarification to show the role of colors.
>
> See Section 3.1, Figure 2 and 3rd paragraph.
>
> **Q2 (same question from c5nB, dUff, and rxHq):**
>
> This distinction — tabular models relying on exact, discrete structures versus deep learning models relying on continuous approximations — is crucial. Although largely open for future work, we should highlight a few directions.
>
> See Section 5 - Extending Design Principles to Deep RL.
>
> **Q3 (same question from Hujr and dUff):**
>
> Figures 7a and 7b illustrate the effects of variance. Figure 7a shows the variability of Naive Policy Selection across 20 seeds, while the Non-stationary and Stationary methods remain stable. Figure 7b shows that the hypervolume exhibits minimal variance.
>
> We plan to include more detailed variance analyses in future revisions.

---

### Official Review · Reviewer_Hujr · 2025-11-01

**Soundness:** 2
**Presentation:** 2
**Contribution:** 1
**Rating:** 2
**Confidence:** 5

**Summary:**

This paper presents a trajectory-centric framework for Temporal-Difference (TD)-based multi-policy MORL that aims to achieve stable and interpretable behavior in infinite-horizon settings.

Unlike prior deep multi-policy approaches such as Pareto Conditioned Networks (PCN), which depend on supervised retraining and curated data, the proposed framework incrementally learns multiple Pareto-optimal policies through TD updates.

The authors introduce a set of design principles—including trajectory-level policy tracking, unification of stationary and non-stationary policies, removal of spurious domination, and explicit cycle detection—to ensure reliable policy following and meaningful undiscounted returns.

Through eight ablation studies on the DeepSeaTreasure benchmark, each design element’s contribution to policy stability, interpretability, and diversity is analyzed.

The work positions itself as a conceptual and algorithmic foundation for future TD-based extensions to deep multi-objective reinforcement learning.

**Strengths:**

- The paper clearly articulates the gap between supervised deep MORL (e.g., PCN) and online TD-based methods and proposes a principled bridge through trajectory-centric design.
- The trajectory-level policy-following mechanism (color labeling and Policy-Transition Table) provides a novel and interpretable way to maintain consistency across multiple Pareto-optimal policies.
- The framework’s unified handling of stationary and non-stationary policies offers full Pareto-front recovery while maintaining predictable policy behavior.
- The ablation results convincingly demonstrate the functional contribution of key components such as spurious domination correction, SSM, and cycle detection to stability and interpretability.

**Weaknesses:**

- The paper does not compare against established baselines such as PCN, MPQ-Learning, or Pareto-DQN, all of which include experiments on broader environments (e.g., Minecraft-based or continuous-control benchmarks).
- The evaluation is limited to the tabular DeepSeaTreasure environment, so scalability and generalization remain untested.
- The theoretical grounding of the proposed principles is mostly heuristic; there is no formal convergence or optimality analysis supporting the modifications.
- Experimental justification is largely qualitative, relying on hypervolume and visual coverage metrics rather than statistical performance comparisons.
- Algorithmic structure is complex (color labeling, PTT, SSM, cycle detection), potentially limiting reproducibility and computational efficiency.
- As a result, while the framework is conceptually interesting, it lacks both the theoretical justification and large-scale empirical evidence needed to confirm its practical advantage.

**Questions:**

The proposed framework introduces several heuristic yet intuitively reasonable design principles (e.g., trajectory coloring, stationary-segment mapping, and spurious domination correction).

While the motivation behind each component is clear, it remains uncertain why these heuristics consistently lead to better learning dynamics.

Could the authors provide theoretical justification or empirical evidence—beyond ablation comparisons—that explains why these mechanisms are effective or under what conditions they provably improve convergence or Pareto-front coverage?

---

> ### Author Response · Authors · 2025-11-23
>
> First, thank you for your thoughtful and specific questions — we appreciate your feedback.
>
> **W1 (same question from c5nB, Hujr, and rxHq):**
>
> Deep RL approaches are not directly comparable, as they belong to a different class of methods; accordingly, we position our work as addressing a gap in tabular approaches, with the goal that the resulting design principles could inspire Deep RL-based extensions.
>
> At the same time, existing tabular approaches do not apply to the infinite-horizon setting. The algorithms in this work are a natural yet non-trivial extension of these methods to this setting; therefore, adapting prior tabular baselines or comparing to them “as is” would not yield a meaningful comparison. Instead, we performed a series of ablations to validate the specific design choices in our framework.
>
> We’ve added clarification in Section 4, 1st paragraph.
>
> **W2 (same question from c5nB, Hujr, and dUff):**
>
> Classical single-objective tabular methods already struggle to scale to real-world problems, and the multi-policy setting exacerbates this challenge. We discuss algorithmic complexity in the appendix: for non-stationary policies, complexity is dominated by pairwise comparisons to filter non-dominated returns, while for stationary policies, it is limited by cycle detection.
>
> **W3 & W6 & Q1 (same question from c5nB and Hujr):**
>
> Our method simply maps standard Q-learning to the multi-policy setting through a trajectory-centric formulation. The mechanisms we introduce address representational issues that arise when extending TD-style approaches addressing the infinite-horizon setting.  Consequently, convergence is expected to match that of standard (single-policy) tabular Q-learning, as our contributions reorganize — rather than modify — the underlying theoretically established concepts.
>
> Sections 3.1 and 3.2 explain how colors enrich the trajectory representation, enabling us to track policies cleanly. Colors are also used to distinguish between stationary and non-stationary policies: in the non-stationary case, they act as identifiers linking trajectory segments, whereas in the stationary case, they also enforce consistency. Conceptually, we decompose elements of a single policy representation to enable the coexistence of multiple policies in the Q-table and PTT. Section 3.3 identifies cases of spurious domination that arise when trajectory information is not preserved. We address this by reformulating how trajectory length and average reward are stored and used. In Section 3.4, we discuss how the same constructs allow us to preserve the total (undiscounted) return: again, by reorganizing the information in the Average-Return and Q-functions rather than introducing new quantities. Finally, Sections 3.5 and 3.6 complement Section 3.3 by detailing how to handle the stationary case and how to preserve the structural requirements specific to it.
>
> Our ablations provide empirical evidence in the DeepSeaTreasure problem, although solving different tasks would further strengthen our case.
>
>
> **W4 (same question from Hujr and dUff):**
>
> Figures 7a and 7b illustrate the effects of variance. Figure 7a shows the variability of Naive Policy Selection across 20 seeds, while the Non-stationary and Stationary methods remain stable. Figure 7b shows that the hypervolume exhibits minimal variance.
>
> We plan to include more detailed variance analyses in future revisions.
>
> **W5:**
>
> While we will continue to refine the presentation, we believe each proposed component is well-motivated and clearly justified. Addressing the identified issues requires these components to work together, which we view as evidence of the thoroughness of our investigation. For completeness, we have uploaded the code as supplementary material, and the appendix includes the full algorithm for reproducibility, along with the complexity analysis. Although the resulting method may appear computationally expensive, this is a natural consequence of the multi-policy setting: for non-stationary policies, complexity is driven by filtering non-dominated returns, while for stationary policies it stems from the need to detect cycles. At the same time, our solution and implementation contribute to strong sample efficiency.

---

### Official Review · Reviewer_c5nB · 2025-11-01

**Soundness:** 2
**Presentation:** 2
**Contribution:** 2
**Rating:** 2
**Confidence:** 5

**Summary:**

The paper studies multi-objective RL in the infinite-horizon regime and argues that TD-style, online updates can support learning and executing a set of Pareto policies when paired with specific design prescriptions. The method is trajectory-centric: it attaches color labels to transitions and uses a Policy-Transition Table to keep an agent on a chosen Pareto policy, handles both stationary and non-stationary behaviors, normalizes away length/frequency biases that cause spurious dominance, and detects/encapsulates cycles so undiscounted averages remain meaningful. Experiments are ablations on DeepSeaTreasure (MO-Gym) that isolate the effect of each ingredient rather than contrasting against external baselines.

**Strengths:**

1. The paper identifies the missing link between deep supervised MORL and TD-based online methods and proposes a structured framework to bridge them.
2. The trajectory-centric policy-following mechanism offers a concrete and interpretable way to maintain consistent Pareto policies during learning.
3. Integrating stationary and non-stationary policy handling allows comprehensive Pareto-front reconstruction under infinite-horizon settings.
4. The ablation analysis effectively isolates how each design choice (e.g., SSM, cycle detection) contributes to policy stability and learning reliability.

**Weaknesses:**

- The paper does not compare against established baselines such as PCN, MPQ-Learning, or Pareto-DQN, all of which include experiments on broader environments (e.g., Minecraft-based or continuous-control benchmarks).
- The evaluation is limited to the tabular DeepSeaTreasure environment, so scalability and generalization remain untested.
- The theoretical grounding of the proposed principles is mostly heuristic; there is no formal convergence or optimality analysis supporting the modifications.
- While the framework is conceptually interesting, it lacks both the theoretical justification and large-scale empirical evidence needed to confirm its practical advantage.

**Questions:**

1. The proposed principles are largely intuitive and heuristic. Can you clarify why they work in practice? In particular, could you provide theoretical analysis or empirical evidence beyond ablations that explains their actual effectiveness?
2. How do the bias-correction terms (trajectory length, reward frequency) theoretically influence convergence or Pareto coverage?
3. Compared to existing MORL baselines (PCN, MPQ-Learning), how do you expect the proposed framework to scale to larger or continuous domains such as Minecraft or complex control environments?

---

> ### Author Response · Authors · 2025-11-23
>
> First, thank you for your thoughtful and specific questions — we appreciate your feedback.
>
>
> **W1 (same question from c5nB, Hujr, and rxHq):**
>
> Deep RL approaches are not directly comparable, as they belong to a different class of methods; accordingly, we position our work as addressing a gap in tabular approaches, with the goal that the resulting design principles could inspire Deep RL-based extensions.
>
> At the same time, existing tabular approaches do not apply to the infinite-horizon setting. The algorithms in this work are a natural yet non-trivial extension of these methods to this setting; therefore, adapting prior tabular baselines or comparing to them “as is” would not yield a meaningful comparison. Instead, we performed a series of ablations to validate the specific design choices in our framework.
>
> We’ve added clarification in Section 4, 1st paragraph.
>
> **W2 (same question from c5nB, Hujr, and dUff):**
>
> Classical single-objective tabular methods already struggle to scale to real-world problems, and the multi-policy setting exacerbates this challenge. We discuss algorithmic complexity in the appendix: for non-stationary policies, complexity is dominated by pairwise comparisons to filter non-dominated returns, while for stationary policies, it is limited by cycle detection.
>
> **W3 & W4 & Q1 & Q2 (same question from c5nB and Hujr):**
>
> Our method simply maps standard Q-learning to the multi-policy setting through a trajectory-centric formulation. The mechanisms we introduce address representational issues that arise when extending TD-style approaches addressing the infinite-horizon setting.  Consequently, convergence is expected to match that of standard (single-policy) tabular Q-learning, as our contributions reorganize — rather than modify — the underlying theoretically established concepts.
>
> Sections 3.1 and 3.2 explain how colors enrich the trajectory representation, enabling us to track policies cleanly. Colors are also used to distinguish between stationary and non-stationary policies: in the non-stationary case, they act as identifiers linking trajectory segments, whereas in the stationary case, they also enforce consistency. Conceptually, we decompose elements of a single policy representation to enable the coexistence of multiple policies in the Q-table and PTT. Section 3.3 identifies cases of spurious domination that arise when trajectory information is not preserved. We address this by reformulating how trajectory length and average reward are stored and used. In Section 3.4, we discuss how the same constructs allow us to preserve the total (undiscounted) return: again, by reorganizing the information in the Average-Return and Q-functions rather than introducing new quantities. Finally, Sections 3.5 and 3.6 complement Section 3.3 by detailing how to handle the stationary case and how to preserve the structural requirements specific to it.
>
> Our ablations provide empirical evidence in the DeepSeaTreasure problem, although solving different tasks would further strengthen our case.
>
>
> **Q3 (same question from c5nB, dUff, and rxHq):**
>
> This distinction — tabular models relying on exact, discrete structures versus deep learning models relying on continuous approximations — is crucial. Although largely open for future work, we should highlight a few directions.
>
> See Section 5 - Extending Design Principles to Deep RL.

---

### Note · Authors · 2026-01-20

I have read and agree with the venue's withdrawal policy on behalf of myself and my co-authors.